# Neurofilaments: The C-Reactive Protein of Neurology

**DOI:** 10.3390/brainsci10010056

**Published:** 2020-01-18

**Authors:** Kate L. Lambertsen, Catarina B. Soares, David Gaist, Helle H. Nielsen

**Affiliations:** 1Department of Neurology, Odense University Hospital, J.B. Winsloewsvej 4, 5000 Odense C, Denmark; klambertsen@health.sdu.dk (K.L.L.); catarinamsoares@hotmail.com (C.B.S.); david.gaist@rsyd.dk (D.G.); 2Department of Neurobiology Research, Institute of Molecular Medicine, University of Southern Denmark, J.B. Winsloewsvej 21, st, 5000 Odense C, Denmark; 3BRIDGE—Brain Research—Inter Disciplinary Guided Excellence, Department of Clinical Research, University of Southern Denmark, J.B. Winsloewsvej 19, 3. sal, 5000 Odense C, Denmark; 4Department of Clinical Research, Neurology Research Unit, Faculty of Health Sciences, University of Southern Denmark, Campusvej 55, 5230 Odense, Denmark

**Keywords:** neuroaxonal damage, biomarker, blood, serum, plasma

## Abstract

Neurofilaments (NFs) are quickly becoming the biomarkers of choice in the field of neurology, suggesting their use as an unspecific screening marker, much like the use of elevated plasma C-reactive protein (CRP) in other fields. With sensitive techniques being readily available, evidence is growing regarding the diagnostic and prognostic value of NFs in many neurological disorders. Here, we review the latest literature on the structure and function of NFs and report the strengths and pitfalls of NFs as markers of neurodegeneration in the context of neurological diseases of the central and peripheral nervous systems.

## 1. Introduction

The interest in neurofilaments (NFs) and their role as disease biomarkers has grown immensely in recent years. NFs were first tested as possible biomarkers by Rosengren et al., who detected an increase in cerebrospinal fluid (CSF) NF-light chain (NF-L) in patients with Alzheimer’s disease (AD) and amyotrophic lateral sclerosis (ALS) compared to controls [1]. This study sparked the interest in NFs as biomarkers, especially the NF-L subunit, not only in neurodegenerative diseases but also in inflammatory, vascular, and traumatic diseases of the central nervous system (CNS) and recently in the peripheral neuropathies (Table 1). Since NFs are solely located in the neuronal cytoskeleton and are released to the interstitial fluid in high quantities upon axonal injury and/or neurodegeneration, they are highly specific for neuronal damage and death [2]. From the interstitial fluid, NFs move to the CSF and subsequently to the blood, where their levels can be measured in serum and plasma and correlated to the extent of axonal damage or neurodegeneration (Figure 1 and Table 1). In recent years, CSF has been the main source for NF analysis, but this is obtained infrequently and only in clinically relevant pathologies due to the invasive nature of lumbar punctures. This has meant that most studies using NF as a marker were restricted to a small spectrum of CNS diseases and disregarded the axonal pathologies of the peripheral nervous system (PNS), not only because of limitations in obtaining relevant biological material like CSF but also because of the quantification methods available. Techniques like ELISA and electrochemiluminescence (ECL) assays can detect NF in peripheral blood, but their sensitivity does not allow very reliable measurements or identification of small variations in concentrations. Single molecule array technology (Simoa™) was introduced in 2010 [3], however, and its higher sensitivity has greatly helped in establishing NF-L as a biomarker in CSF, serum, and plasma [4,5]. Here, we review fundamental developments in the assessment of NFs as biomarkers in human studies and the contribution of NFs to disease monitoring and effectiveness of current and developing therapies.

## 2. The Structure and Function of Neurofilament

NFs are present in the neuronal cytoskeleton, especially the axon, and are type IV intermediate filaments with a diameter of approximately 10 nm. These proteins are mostly found in adult myelinated neurons of the CNS and large-caliber myelinated axons of the PNS. NFs have several functions but are mainly responsible for increasing and maintaining axonal caliber and therefore improving relay of electrical impulses along the axons. Human NF is composed of four subunits: NF-L, neurofilament medium chain (NF-M), and neurofilament heavy chain (NF-H) with molecular weights of approximately 68, 160, and 205 kDa, respectively; the fourth subunit will depend on the location of the protein, therefore α-internexin (66 kDa), mostly present in spinal cord and optic nerve, is found in the CNS while peripherin (58 kDa), present in root ganglia neurons, is located in the PNS. NF chains are comprised of a non-helical amino terminal ‘head’ domain, a central α-helical ‘rod’ domain, and a variable length carboxyl terminal ‘tail’ domain [73,74,75,76]. Interference in NF assembly, metabolism, and release due to axonal injury is related to several neurodegenerative diseases, which is further elaborated on in this review.

## 3. Neurofilament in Neurological Disease

### 3.1. Peripheral Neuropathy

Peripheral neuropathy is a broad group of diseases that share the common feature of damage to peripheral nerves. Depending on the nerves affected, symptoms may include autonomic dysfunction, impaired sensation, or impaired movement. Onset of peripheral neuropathy can be acute, subacute, or chronic. Most peripheral neuropathies are acquired due to a large variety of underlying causes ranging from diabetes and alcohol abuse to inflammation and infection. A minority of peripheral neuropathies are hereditary.

NFs have mostly been associated with the hereditary neuropathies like Charcot–Marie–Tooth (CMT), in which several subtypes have been shown to include genetic mutations in the *NEFL* gene, leading to disruption of NF assembly and transport, and abnormal NF accumulation [77,78,79,80,81]. It is therefore not surprising that plasma NF-L is increased in CMT and correlates with disease severity in several subtypes [13].

The acquired neuropathies have also been the focus of NF studies. Serum NF-L is elevated in polyneuropathies as diverse as chronic inflammatory demyelinating polyneuropathy, anti-MAG neuropathy, and vasculitic neuropathy [10,11]. Serum levels were not only elevated compared to healthy controls but also seemed to decline with remission [11] and to correlate with disease severity and outcome [10].

In the case of Guillain–Barré syndrome (GBS), a rapidly evolving immune-mediated demyelinating polyradiculoneuropathy, axonal involvement is a well-known indicator of poor prognosis [82]. In line with this, CSF NF-L was not only elevated in the acute phase and correlated with disability [2] but was also a strong predictor of persistent disability [12]. These results were recently confirmed by plasma analyses [2,10].

Taken together, these studies suggest that NF-L might be a promising biomarker of disease and outcome in neuropathies. Studies of CMT often include large cohorts and sensitive detection methods. Although NF-L might have a role as a prognostic biomarker, its diagnostic sensitivity and specificity are most likely inferior to genetic testing.

In the acquired neuropathies, special care must be taken when interpreting results as studies are still limited by sample size and differences in detection technologies (Table 1). Further studies using reliable reproducible detection methods, like the Simoa technology, on larger cohorts are needed before final conclusions can be made.

### 3.2. Motor Neuron Disease

Motor neuron diseases (MNDs) are a group of neurodegenerative disorders characterized by degeneration of motor neurons. They comprise a spectrum of clinically defined diseases with involvement of upper and/or lower motor neurons and include progressive bulbar palsy, progressive muscular atrophy, primary lateral sclerosis, and some rare variants. The most common form of MND is ALS, a relentlessly progressive and ultimately lethal condition caused by degradation of lower and upper motor neurons in the motor cortex and spinal cord. Although most cases are sporadic, 5%–10% show a clear familial accumulation with a Mendelian pattern [83]. The diagnosis is made by classic clinical findings of progressive muscular paralysis, atrophy, fasciculations, and hyperreflexia, supported by electrophysiological findings. To better understand and predict the prognosis and to assist in the development of treatments, biomarkers in the blood and CSF have been intensely investigated.

Both NF-L and phosphorylated NF-H have been studied as possible biomarkers for ALS [14,16,84,85]. Serum and plasma NF-L levels, tested using an ECL immunoassay, showed higher levels in ALS subjects compared to healthy controls as well as increased NF-L concentrations in ALS patients with a fast progressing phenotype, stabilizing over time [14]. Even in the early stages, where symptoms are still minor, NF-L in blood and CSF can distinguish ALS patients from healthy controls, but more importantly can also distinguish ALS from other MNDs with similar features but with much better prognosis [14,15,86,87].

Furthermore, high NF-L levels in both CSF and blood and NF-H in CSF were found in symptomatic ALS patients compared to asymptomatic genetic mutation carriers, whose levels were similar to healthy controls [16]. Interestingly, an increase in blood NF-L was observed in asymptomatic individuals as early as 12 months before symptom development, suggesting that neurodegeneration precedes symptom development [16].

As a prognostic marker, CSF NF-L levels are lower in patients with slower disease progression [88], and both serum and CSF NF-L as well as NF-H levels are associated with the number of muscular regions involved [89]. Similarly, serum and CSF NF-L levels are high in patients with rapid progression and shorter overall survival [17,85].

Apart from symptomatic treatment, disease-modifying therapies in MNDs are very limited, with riluzole being the only approved pharmacological therapy for ALS. A single study showed no difference in serum NF-H of patients on riluzole [84], but experiences from other diseases suggest that NFs might also be attractive candidates as endpoints in clinical trials in MNDs.

The studies on MNDs are generally solid, with large number of patients compared to appropriate controls (Table 1), and they suggest a role for NF-L as a diagnostic and prognostic marker, especially regarding measurements in the CSF. While measurements of NF-L in plasma or serum might be attractive, they must be evaluated with care and especially if less sensitive detection methods are applied. Considering the severity of these diseases, it is important to stress that NF-L elevations are a general feature of neurodegeneration and not specific to MND, thus results must be interpreted in the clinical context.

### 3.3. Multiple Sclerosis

Multiple sclerosis (MS) is a pathology characterized by continuous inflammation of the CNS that leads to demyelination of axons and consequent neurodegeneration. MS is often considered as either a relapsing-remitting disease or as a progressive disease with continuous disability progression, reflecting an ongoing underlying neurodegeneration. Diagnosis is made based on classical symptoms with dissemination in time and space, use of clinical biomarkers such as oligoclonal bands in CSF, magnetic resonance imaging (MRI), and exclusion of other plausible diseases [90,91,92]. In the past decade, however, the emergence of other biomarkers in CSF, like NF-L, has provided further insight into MS diagnosis and disease progression [93,94,95,96]. Several studies have shown that NF-L is elevated in the blood and CSF of newly diagnosed MS patients and that this is correlated with disease severity and prognosis [21,97,98,99]. NF-L is also elevated during disease activity such as a clinical relapse of new lesions on MRI [100,101]. This makes NF-L an attractive diagnostic biomarker in MS although its specificity does not allow differentiation from MS mimics like neuromyelitis optica spectrum disorders or other neuroinflammatory disorders [102,103].

Repeated NF-L measures in the blood might be an attractive marker of treatment response since treatment with disease-modifying drugs such as dimethyl fumerate, interferon-beta, natalizumab, fingolimod, cladribine, and alemtuzumab resulted in significant reductions in NF-L in the CSF and blood [21,97,104,105,106,107].

Brain atrophy is also a well-known hallmark of the neurodegeneration seen in MS. It is therefore interesting that even early measures of blood NF-L in newly diagnosed MS patients can predict brain atrophy [22] and lesion load on MRI [23], probably reflecting high disease activity since this can be modified by initiation of effective treatment [108]. However, NF-L levels have shown poor correlation to disability scores such as the Extended Disability Status Scale (EDSS) [22,23,24,93,109,110]. This is most likely due to the emphasis on gait performance in the EDSS that may overestimate the influence of spinal cord lesions over cerebral lesions, which can be extensive even at low EDSS scores. Whether NF-L levels also correlate with cognitive measures and fatigue in MS remains to be confirmed although recent studies seem to suggest a negative correlation [23,111,112,113].

The study of NF-L is a rapidly growing field in MS, and studies are moving from CSF measurements to serum/plasma measurements using sensitive techniques. As a diagnostic marker, NF-L might not provide much additional value to the already applied diagnostics as its specificity does not allow differentiation from MS mimics. However, as a marker of disease activity and treatment response, repeated NF-L measurements in blood are moving towards an established biomarker with solid studies on large cohorts under various medications using sensitive techniques (Table 1). The focus has so far been on the relapsing-remitting phenotype where treatment options are many. However, NF-L might also be an attractive marker of disease activity in the progressive forms of MS, where neurodegeneration outweighs neuroinflammation, and it might provide assistance in clinical trials of new medications.

### 3.4. Alzheimer’s Disease

The neurodegenerative disorder of Alzheimer’s disease (AD) is the most frequent cause of dementia, accounting for 60%–80% of all cases [114]. The condition has many clinical features, the most essential being loss of memory. AD pathology is characterized by amyloid plaques originating from the amyloid precursor protein (APP) metabolism. Amyloid plaques in AD are abnormally folded Aβ40 or Aβ42 and are deposited extracellularly. Neurofibrillary tangles, which are also prominent in AD pathology, are made up of paired helical filaments composed of hyperphosphorylated tau, a microtubule-stabilizing protein. AD’s pathological processes lead to neurodegeneration and inflammation, causing neuronal and synaptic loss and progressive macroscopic atrophy [115,116]. CSF and blood levels of tau, phospho-tau, and Aβ_1-42_ are used to solidify the diagnosis and to differentiate between AD and other forms of dementia [117,118,119,120]. With the complications associated with using CSF, several recent studies have turned to other proteins found in blood, like NF-L, to investigate their role in disease progression and neurodegeneration [27,28,30]. Most studies have revealed increased NF-L concentrations in patients with AD and other forms of dementia compared to healthy controls (Table 1). A particular study also noted that high NF-L levels in plasma of AD patients correlated with poor cognition and brain atrophy [30]. NF-L has also been used as a biomarker for cognition impairment, revealing that high NF-L plasma levels correlate with impaired cognition in AD, giving potential for this protein to be a biomarker for mental decline [31]. Down syndrome (DS) individuals have an increased risk of developing early-onset AD, with approximately two-thirds of all individuals being affected by this form of dementia [121]. NF-L plasma levels were indeed found to be higher in DS and seemed to increase with age and to predict dementia status [122]. A cross-sectional study on DS concluded that NF-L was useful in diagnosing AD in DS individuals and outperformed other fluid biomarkers [29]. All these studies mentioned used Simoa technology and concluded that NF-L is a potential non-invasive biomarker for AD, helping to solidify a diagnosis and monitor disease progression.

Given that cognitive impairment and particularly frontotemporal dementia (FTD) can be found in up to 10%–15% of patients with ALS [123], it is interesting that in this particular form of dementia, blood and CSF NF-L levels are associated with functional outcome, brain atrophy, and severity, and can be used to discern FTD from other types of dementia and healthy controls but not be used to distinguish between FTD subtypes [124,125,126]. However, the use of varied biomarker panels comprised of NF-L, total and phosphorylated tau, and Aβ_1-42_ improved discrimination between AD dementia, FTD, and some other types of dementia [118,127,128] and increased sensitivity in AD stage differentiation using NF-L and fatty acid binding protein 3 [129]. A panel of biomarkers that includes proteins involved in the pathogenic mechanisms of dementia helps improve diagnosis and clinical staging of the disease.

Taken together, the evidence so far supports NF-L in the CSF and blood as a diagnostic marker of dementia, with AD being most frequently investigated (Table 1). As the hallmark of dementia is ongoing neurodegeneration, it is not surprising that NFs are elevated. The specificity of NF-L is therefore also limited and is strongest as part of a varied biomarker panel. Especially in the aged population, data must be interpreted in the clinical context due to age-dependent variations in NF-L levels, which calls for special attention to the age-matched healthy controls provided in many of the studies listed in Table 1.

### 3.5. Huntington’s Disease

Huntington’s disease (HD) is a genetic neurodegenerative pathology where a CGA triplet repeat expansion on the huntingtin gene (*HTT*) originates as an expanded polyglutamine segment in the huntingtin protein. The expanded protein forms intranuclear aggregates toxic to neuronal cells, causing neuronal dysfunction and cell death. In HD, biomarkers have the potential role of assessing therapeutic efficacy, and most recently NF has been tested in both CSF and plasma [35,36,130,131] (Table 1). A recent study investigated the correlation between mutant huntingtin (mHTT) and CSF and plasma NF-L levels in HD patients, revealing NF-L to be a clinically stronger marker than mHTT, even within the study’s limitations [33]. ELISA showed that NF-L levels in CSF of HD patients were significantly higher than those in matched controls [36], while plasma levels of NF-H, using ELISA, excluded this NF subunit as a potential biomarker [130]. Most recently, a cohort analysis using highly sensitive Simoa showed great potential of plasma NF-L as a prognostic marker and as a timepoint reflection for motor and cognitive impairment [35]. The latest generation of quantitative methods could enable the use of NF-L as a stronger observational and therapeutic biomarker for HD. Genetic therapies combined with NF-L observation could be important tools in the treatment and monitoring of this pathology.

Studies on HD are limited and have small samples sizes, however, most likely due to the low prevalence of the disease. Further studies using sensitive techniques are therefore needed before conclusions can be made, especially considering measurements on peripheral blood. However, while the diagnostic value of NF-L is probably minor compared to genetic testing, NF-L might be a promising prognostic marker and may assist in the development of therapies by acting as a marker in clinical trials.

### 3.6. Parkinson’s Disease and Parkinsonian Disorders

Atypical parkinsonian disorders (APDs) such as multiple system atrophy (MSA), progressive supranuclear palsy (PSP), and corticobasal degeneration (CBG) often present with similar and overlapping symptomatology as Parkinson’s disease (PD), especially in early disease stages. CSF NF-L is a promising marker to separate PD from APD, as NF-L is increased in APD compared to PD [132,133] and can be used to discriminate between APD and PD with a high degree of accuracy [134,135,136] (Table 1). Blood NF-L has also recently been demonstrated to be a promising diagnostic marker to separate PD from APD, with significantly higher blood NF-L levels in APD patients compared to PD patients and healthy controls [38,137]. As NF-L levels in PD patients are comparable to healthy controls, NF-L appears to be a better prognostic marker for APD than for PD. Given that NF-L is a marker of large myelinated axons, it is possible that axonal degeneration is less severe in PD than in APD so that blood NF-L does not increase in PD compared to healthy controls. This is in line with MRI diffusion studies demonstrating extensive white matter injury in APD but not in PD [138,139,140] and studies demonstrating that NF-L levels correlate with disease severity in APD [38,136,141].

As PD and APD are still largely clinical diagnoses, supportive biomarkers are highly sought. Evidence suggests that at least in the CSF, NF-L might be a strong candidate as a discriminating diagnostic biomarker. Sensitivity is diminished, however, when testing in the peripheral blood, stressing the need for reliable and sensitive detection methods and preferably Simoa technology (Table 1). The parkinsonian disorders show great variation in their clinical presentation, with various degrees of clinical symptoms including dementia. As an ongoing progressive disease, this calls for special attention in the matching of study populations with healthy controls and is often the weak point in studies of less common diseases with multiple subtypes. Further studies on larger cohorts are therefore required.

### 3.7. Stroke

Stroke is a leading cause of death and disability worldwide, making this condition a major health concern [142,143]. Neuronal death caused by deprivation of glucose and oxygen to the affected area leads to the release of axonal proteins, like NF. This protein has a potential for being a biomarker for stroke severity and post-stroke outcome. Studies have shown how the CSF NF-H levels, measured using ELISA, increased in aneurysmal hemorrhagic and acute ischemic stroke and related to patient outcome [55] (Table 1). One study assessed CSF NF-L levels along with a panel of other neuronal biomarkers that were measured in acute ischemic stroke patients, and found higher levels of this NF-L compared to controls as well as a correlation with the degree of white matter hyperintensities [46]. However, the emergence of more sensitive detection techniques has allowed the detection of NF-L in the peripheral blood. Serum NF-L levels, analyzed using Simoa, of patients with MRI-confirmed small subcortical infarcts were higher than in healthy controls [49] (Table 1). This trend continued at three months follow up, but NF-L levels seemed to normalize fifteen months post-stroke [49], indicating that NF-L could be a tool for monitoring infarct extent after a stroke. NF-L serum levels are higher in ischemic stroke cases than in healthy controls, while also differentiating between ischemic stroke and transient ischemic attack (Table 1). Higher NF-L levels were associated with more severe disability scores [44,45]. In a large prospective study, Pedersen et al. used Simoa to study the association between functional outcome and serum NF-L levels sampled within the first 1–14 days, 3 months, and 7 years after ischemic stroke [41]. The authors showed that serum NF-L levels increased with time between stroke and blood sampling and observed the highest concentrations at 3 months post-stroke. Serum NF-L levels correlated with functional outcome at all timepoints investigated although the strongest correlation was between functional outcome and NF-L levels sampled 3 months after stroke onset. Importantly, for all main etiological stroke subtypes, both acute serum NF-L and 3 month levels were significantly higher in stroke patients than in controls. The release of NF-L after acute neuronal damage could be due to continuous breakdown of the blood–brain barrier, but persistent post-ischemic inflammatory and immunological processes could also explain lengthened NF-L release [4]. Further studies on this research topic could help us understand the role of inflammation in stroke and axonal injury.

As stroke is a common disease, study populations are often large with well-matched control populations, providing strong evidence for NF-L as a diagnostic and prognostic marker. In contrast to the chronic diseases, stroke represents an acute neurodegeneration followed by a recovery phase with little or no ongoing neuronal pathology. NF-L levels are therefore likely to be time-dependent, with decreasing levels over time. This makes comparison between studies enrolling from 1 to 14 days post-stroke difficult, even when using the same sensitive and reliable detection method (Table 1). Since very little is known about the dynamics of NF-L release after an acute pathology, this must be investigated further. An appropriate consensus time for measurement must be established before NF-L can be considered a reliable prognostic and diagnostic biomarker of stroke.

### 3.8. Traumatic Axonal Injury

Traumatic brain injury (TBI) is the principal cause of death and morbidity in individuals under 45 years old and is a topic of high concern [144]. TBI can be classified as mild, moderate, or severe, typically using the Glasgow Coma Scale (GCS) [145,146,147], with mild traumatic brain injury (GCS score 14–15) being the most common. There is a need for biomarkers that can assess severity and outcome in TBI as well as the risk of developing chronic traumatic encephalopathy due to repeated exposure to head trauma [59,60].

Being an unspecific marker of neurodegeneration and axonal damage, NF-L is a promising marker of injury in TBI (Table 1). In severe TBI, both CSF and serum NF-L sharply increase over the first 2 weeks compared to healthy controls and appear to be reliable predictors of poor outcome [56,57,58]. Similar predictions may be possible from serum NF-L in mild traumatic injury although care must be taken when patients are over 60 years old or have preexisting neurological disorders [59,60].

Most studies in mild TBI or concussion have focused on athletes in contact sports like boxing, hockey, or American football (Table 1). Uncomplicated concussion in soccer [148] or even American football and hockey [61] does not seem to affect NF-L in serum or CSF. Elevated levels of CSF and serum NF-L have been documented in boxers, however, especially if the trauma incurred during a fight resulted in measurable impact on the GCS [62] or was due to several blows to the head [149]. Furthermore, NF-L stayed elevated for a prolonged time [150] and was still elevated after 3 months, a clear predictor of post-concussion symptoms [58].

Regardless of the type of contact sport, there seems to be a relationship between NF-L levels and the frequency and magnitude of the head impact [58,61,62,63,64,151,152]. It is not yet clear, however, whether NF-L may serve as a biomarker for prediction of when it is safe to return to contact sports without the risk of permanent disability and chronic traumatic encephalopathy [153]. A potential clinical use for this biomarker would be to help clinicians decide whether a patient with TBI should undergo a head CT or MRI in the acute setting. Since the fluid dynamics of plasma and serum NF-L are not completely clear at this time, it is most likely to be used as part of a multipanel array [154].

Similar to TBI, a few studies of traumatic spinal cord injury have demonstrated increased NF-L values in both CSF [65] and serum [66], where early elevations correlated with motor outcome 3–12 months after trauma. However, this area is still largely unexplored.

As TBI is an area of keen interest, studies of NF-L as a prognostic marker are many and often include large patient cohorts (Table 1). The evidence strongly supports NF-L as a prognostic marker in severe TBI although very little is known about the dynamics of NF-L in the recovery period, and the optimal timepoint for measurement is yet to be established. Studies in mild TBI are often limited by smaller study populations and varying types of head trauma. Nevertheless, the focus on head injury in contact sports provides good opportunities for further studies on large cohorts of healthy young adults with similar trauma. With the growing use of Simoa technology, longitudinal studies of NF-L in peripheral blood may soon provide the necessary evidence to support the use of NF-L as a prognostic marker.

### 3.9. Cardiac Arrest

Cardiac arrest often leads to global cerebral ischemia, and irreversible brain damage is likely to occur by nine minutes after the cessation of blood flow to the brain. The prognosis of out-of-hospital cardiac arrest is poor, with less than 10% survival [155]. Prognostication of brain damage in patients surviving cardiac arrest currently relies mainly on clinical observations, electroencephalography, somatosensory evoked potential, and neuroimaging. However, CSF biomarkers such as NFs have proved valuable as predictors of brain damage and outcome after cardiac arrest [67,68] (Table 1). CSF NF-L levels are increased in cardiac arrest patients 2–3 weeks after the arrest compared to healthy controls, and NF-L levels are significantly higher in patients with poor outcome compared to those with good outcome according to the Glasgow Outcome Scale, activities of daily living, and mini-mental state examination [68]. Plasma [71] and serum [69,72] NF levels have also been measured in cardiac arrest patients (Table 1). In a recent pilot study, Disanto and colleagues [69] used Simoa to detect serum NF-L within 17 days of cardiac arrest and found a positive association between serum NF-L levels and time to return of spontaneous circulation, severity of brain damage estimated by electroencephalogram, and clinical outcome (death status at 1 month). In a large prospective study, Moseby-Knappe et al. [72] used ELISA to establish the potential of serum NF-L as a prognostic marker of outcome after cardiac arrest. Blood samples were collected at 24, 48, and 72 h after return of spontaneous circulation in 717 patients, of whom 360 had a poor neurologic outcome at 6 months. Serum NF-L levels were significantly increased in patients with poor outcome compared to those with good outcome at all timepoints and performed better than the biochemical biomarkers tau, neuron-specific enolase, and S100. The authors concluded that serum NF-L can be used as a predictive marker of long-term poor neurological outcome at 24 h after cardiac arrest.

These findings suggest that NFs can be used as reliable measures of brain damage following cardiac arrest and that blood NF levels are highly predictive of outcome. As this area is relatively unexplored, studies are still scarce and flawed by low patient numbers and variations in the timing of measurement. Future studies will help determine the potential of NFs as future biomarkers after cardiac arrest.

### 3.10. Delirium

Delirium is a common and serious neuropsychiatric syndrome that manifests a new-onset dementia with features of inattention and global cognitive dysfunction. The etiological causes of delirium are many and multifactorial and often reflect ongoing acute medical illness or medical complication. Currently, the diagnosis of delirium is clinically based and depends on the absence or presence of certain features. As delirium appears to be associated with permanent encephalopathy with neuronal damage and/or dysfunction, several studies have investigated NFs as potential new biomarkers of neuroaxonal damage in delirium patients [156,157,158,159,160,161]. In a recent prospective, pilot observational study, Ehler and colleagues [161] demonstrated, using ELISA, that plasma NF-L levels increased in sepsis patients over time and remained stable in patients without sepsis. Furthermore, NF-L levels were higher in patients with sepsis-associated encephalopathy and correlated with functional outcome and death. Elderly hospitalized patients are at high risk of developing delirium and postoperative delirium appears to be associated with increased plasma NF-L levels [159]. The use of NFs as biomarkers in delirium is, however, still controversial as the literature in this field is still scarse. One study demonstrated that serum NF-L, measured using Simoa, increased significantly in patients that developed delirium after cardiac surgery [157], whereas in another study [158] serum NF-H, measured using ELISA, could not be used to predict patients at risk of delirium following cardiac surgery. Whether these results reflect differences in the sensitivity of the assay, ELISA vs. Simoa, differences in the NF investigated, NF-L vs. NF-H, or differences in the underlying disease mechanisms, remain to be elucidated. More studies are warranted to uncover the potential of NFs as biomarkers in delirium.

## 4. Discussion

NFs are often considered the C-reactive protein (CRP) test of neurology, suggesting its use as an unspecific screening marker of neurodegeneration in the PNS or CNS. However, like the CRP, care must be taken when interpreting results obtained in different studies. First, very little is known about NF levels and dynamics in the normal healthy population. Ideally, large populations of healthy individuals are required to generate normative data for reference use. While most studies compare diseased patients with normal healthy controls, many samples are collected from patients referred to hospital due to a complaint that is later resolved or not caused by neurological disease [97]. Such symptomatic controls may not represent the true levels of NF in the general population. Second, many studies also show an age-dependent correlation with NFs [59,60,98]. This is most likely caused by the well-described physiological age-related brain atrophy [162,163]. Similarly, it has been claimed that the subjects’ sex may affect NF levels [164,165] although this remains to be verified. Third, variations in the source of the tested substance need to be considered. In neurological diseases, NF-L is often tested in CSF, serum, or plasma. While CSF has some obvious advantages in CNS disease, its use is often limited by the invasive nature of the procedure that precludes its use as a frequent longitudinal marker. In general, there are strong correlations between NF-L levels in CSF, serum, and plasma [97,101,166,167], but while specificity and sensitivity are high in CSF due to the higher levels of NF-L, sensitivity is often lower in plasma [101,164,166,168] and even lower in serum [97]. Finally, the test methods also affect the results. NF-L is typically measured using one of three assays. The commercially available ELISA [169] has the advantage of being cheap and is readily available in most laboratories, but the manufacturer does not recommended it for blood analyses because of its low sensitivity. ECL-based assays are sensitive and require only a small sample volume to produce a large range of results [2]. These methods have largely been replaced by the Simoa technology, however, as digital immunoassays can significantly improve sensitivity and deliver reliable and robust data on both CSF, serum, and plasma NF-L, even between different laboratories [97,141].

## 5. Conclusions

The development of extremely sensitive immunoassays has increased the potential role of NF-L as a biomarker through avoiding the collection of CSF and enabling frequent measurement in blood and not just the CSF. These ultrasensitive methods have allowed NF-L monitoring to expand to conditions where CSF collection might not be possible and allow repeated measures in longitudinal studies. Together with other clinical and paraclinical measures, NF-L may be able to contribute to diagnosis, improving accuracy, and differentiating between manifestations and stages of certain diseases. However, just as the CRP is an unspecific test of inflammation and disease activity, NF-L must be interpreted in the clinical context, and no single test or value can be used to rule in or out a specific diagnosis. Without normative data for the general population and with various techniques still being applied, care must be taken when interpreting NF levels as predictors of or biomarkers for neurological diseases. Future studies using sensitive and reproducible techniques are needed to further solidify this protein before fully applying it to clinical practice.

## Figures and Tables

**Figure 1 brainsci-10-00056-f001:**
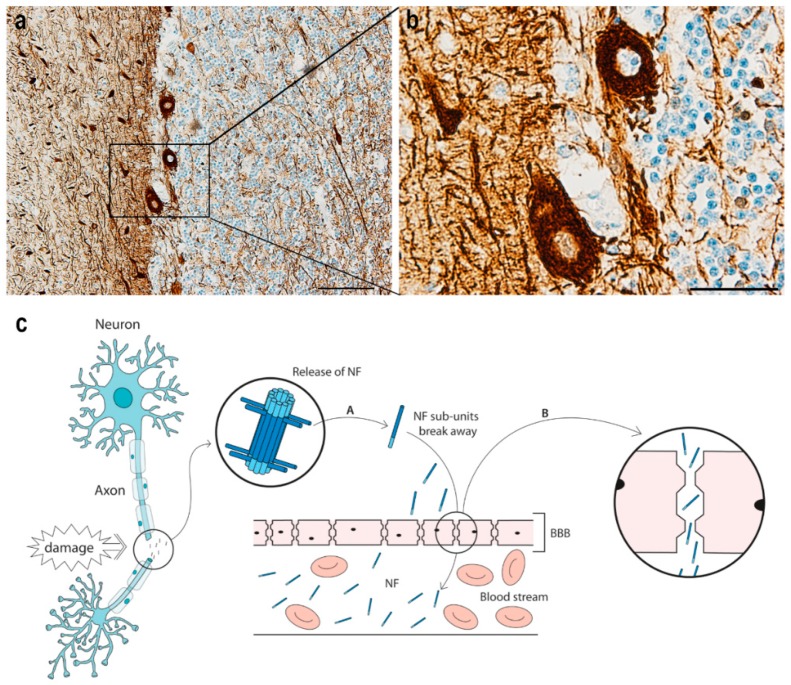
Neurofilament after neuroaxonal damage. (**a**) Immunohistochemical staining of neurofilament-positive neurons in post-mortem human ischemic cerebellum tissue. Scale bar: 100 μm. (**b**) High magnification of squared area in (**a**) showing neurofilament-positive neurons. Scale bar: 40 μm. Neurofilament immunohistochemical staining was performed on parallel tissue sections from post-mortem ischemic brain tissue used in previous studies [6,7,8,9]. Staining was performed using similar protocols and the following antibody: monoclonal mouse anti-neurofilament (phosphorylated and non-phosphorylated NF-H chain) antibody (clone N52, 1:1000, Sigma-Aldrich, St. Louis, MO, USA). The use of human brains was approved by the Danish Biomedical Research Ethical Committee for the Region of Southern Denmark (permission number S-20080042). (**c**) Schematic presentation of neuroaxonal damage leading to neurofilament release. When a neuron and its axon are damaged, neurofilament is released into the extracellular space (A) and subsequently into the cerebrospinal fluid (CSF) and blood (B), where it can be detected in increased levels following neuroaxonal damage. Abbreviations: BBB, blood–brain barrier; NF, neurofilament.

**Table 1 brainsci-10-00056-t001:** Overview of neurofilaments as biomarkers of neurological disease. Abbreviations used in table: AD, Alzheimer’s disease; ADAD, autosomal dominant Alzheimer’s disease; AIS, acute ischemic stroke; ALS, amyotrophic lateral sclerosis; aMCI, amnestic mild cognitive impairment; aSAH, aneurysmal subarachnoid hemorrhage; CA, cardiac arrest; CADASIL, cerebral autosomal-dominant arteriopathy with subcortical infarcts and leukoencephalopathy; CBS, corticobasal syndrome; CeAD, cervical artery dissection; CIDP, chronic inflammatory demyelinating polyneuropathy; CMT, Charcot–Marie–Tooth; CSF, cerebrospinal fluid; ECL, electrochemiluminescence; FTD, frontotemporal dementia; GBS, Guillain–Barré syndrome; GCS, Glasgow Coma Scale; HD, Huntington’s disease; HHT, huntingtin; HS, hemorrhagic stroke; MCI, mild cognitive impairment; MMN, multifocal motor neuropathy; MND, motor neuron disease; MS, multiple sclerosis; MSA, multiple system atrophy; NF-M, neurofilament medium chain; ON, optic neuritis; PD, Parkinson’s disease; pNF-L, plasma neurofilament light chain; p-pNF-H, plasma phosphorylated neurofilament heavy chain; PSP, progressive supranuclear palsy; ROSC, return of spontaneous circulation; RSSI, recent small subcortical infarcts; SAH, subarachnoid hemorrhage; SCI, spinal cord injury; s-pNF-H, serum-phosphorylated neurofilament heavy chain; Simoa, single molecule array; sNF-L, serum neurofilament light chain; SVD, small vessel disease; TIA, transient ischemic attack; WMH, white matter hyperintensities; WML, white matter lesions.

Disease and Sample Size	Protein	Method	Time Profile, Association with Disease Activity, and Diagnostic and/or Prognostic Relevance	Reference
**Neuropathies**				
20 GBS patients and 67 controls	CSF NF-L and sNF-L	ECL	Elevated sNF-L and CSF NF-L in GBS patients compared to controls	[2]
GBS (5), MMN (3), CIDP (12); AntiMAG (3), CIDP + antiMAG (1) vasculitic neuropathy (1) and 25 controls	CSF NF-L; pNF-L	Simoa	Elevated pNF-L and CSF NF-L in GBS, CIDP and antiMAG vs. controls	[10]
30 vasculitic neuropathy patients and 30 controls	sNF-L	Simoa	Elevated sNF-L during active disease vs. controls	[11]
18 GBS and 18 controls	CSF NF-L	ELISA	Elevated CSF NF-L in GBS vs. controls, correlation with severity and outcome	[12]
75 CMT and 67 controls	pNF-L	Simoa	Elevated pNF-L vs. control correlated with severity	[13]
**Amyotrophic Lateral Sclerosis**				
67 controls and 20 ALS patients	CSF NF-L, sNF-L	ECL	Elevated sNF-L and CSF NF-L in ALS vs. controls	[2]
103 patients and 42 controls	CSF NF-L, sNF-L, pNF-L	ECL	Blood-derived NF-L level is an easily accessible biomarker with prognostic value in ALS	[14]
124 ALS, 44 ALS mimics, 65 other neurodegenerative disorders, and 50 healthy controls	sNF-L	Simoa	Serum NF-L is elevated in ALS and can distinguish between ALS mimics and correlate with prognosis	[15]
12 asymptomatic carriers, 64 symptomatic carriers, and 19 healthy family controls	CSF NF-L, sNF-L	Simoa	Symptomatic carriers have higher levels of NFs in serum and CSF compared to controls, blood NF-L increases 12 months before symptom onset	[16]
715 MND, 87 FTD, and 107 controls	CSF NF-L	Simoa	High levels of NF-L associated with shorter survival	[17]
**Multiple Sclerosis**				
56 ON patients and 27 controls enrolled within 28 days of onset (median 16 days)	CSF NF-L	ELISA	No correlation between NF-L and MS-risk parameters	[18]
47 ON patients enrolled within 28 days of onset (median 16 days)	CSF NF-L	ELISA	CSF NF-L predicted visual outcome after ON	[19]
68 ON patients	CSF NF-L	ELISA	CSF NF-L predicted long-term physical and cognitive disability	[20]
589 patients and 33 controls	pNF-L	Simoa	pNF-L levels associated with disease activity and have prognostic value	[21]
74 patients	sNF-L	Simoa	sNF-L correlated with MRI activity	[22]
122 patients	sNF-L	Simoa	sNF-L predicted 10 year lesion load and atrophy	[23]
189 RRMS and 70 PMS	sNF-L	Simoa	sNF-L correlated with concurrent and future clinical and MRI measures of disease activity and severity	[24]
**Alzheimer’s Disease**				
67 controls and 20 AD patients	CSF NF-L, sNF-L	ECL	Elevated sNF-L and CSF NF-L in AD patients compared to controls	[2]
42 ADAD patients (22 symptomatic and 20 asymptomatic mutation carriers) and 18 controls	CSF NF-L, sNF-L	Simoa	sNF-L correlated with clinical and cognitive measures in ADAD and with CSF NF-L. Elevated sNF-L in symptomatic vs. asymptomatic carriers and controls	[25]
243 ADAD mutation carriers and 162 controls	CSF NF-L, sNF-L	Simoa	Elevated sNF-L and CSF NF-L levels in ADAD compared to controls. sNF-L predicted disease progression and neurodegeneration at the early pre-symptomatic stages	[26]
198 aMCI and 187 AD patients, and 193 controls	pNF-L	Simoa	Elevated pNF-L in aMCI and AD patients compared to controls and elevated in AD compared to aMCI. pNF-L associated with cognition	[27]
99 MCI and 33 early AD patients, and 41 controls	pNF-L	Simoa	Elevated pNF-L in MCI and early AD patients compared to controls	[28]
Down syndrome-associated AD (194 asymptomatic, 39 prodromal, and 49 symptomatic), and 67 controls	CSF NF-L, pNF-L	ELISA	Elevated NF-L in prodromal and symptomatic AD compared to controls. pNF-L and CSF NF-L differentiated between asymptomatic, prodromal, and symptomatic AD. CSF NF-L and pNF-L correlated.	[29]
197 MCI and 180 AD patients, and 193 controls	pNF-L	Simoa	Elevated pNF-L in MCI and AD patients compared to controls. CSF NF-L and pNF-L correlated. pNF-L associated with poor cognition, AD-related atrophy, and brain hypometabolism	[30]
56 MCI and 119 AD patients, and 59 controls	pNF-L	Simoa	Elevated pNF-L in AD compared to controls. pNF-L associated with cognition	[31]
**Huntington’s Disease**				
11 premanifest and 12 manifest HD patients	CSF NF-L	ELISA	CSF NF-L correlated with 5 year probability of disease onset, functional capacity, and total motor score	[32]
20 premanifest, 40 manifest HD patients, and 20 controls	CSF NF-L and pNF-L	ELISA	CSF and pNF-L were increased in premanifest and manifest HD patients vs. controls. Manifest HD displayed higher levels vs. premanifest HD patients	[33]
29 early, 29 premanifest, and 30 moderate HD patients, and 29 controls	pNF-H	ELISA	No correlation between pNF-H and disease stage	[34]
201 individuals carrying HHT mutations and 97 controls	pNF-L	Simoa	pNF-L correlated with clinical and MRI findings	[35]
35 HD patients and 35 controls	CSF NF-L	ELISA	Elevated CSF NF-L in HD patients vs. controls. CSF NF-L correlated to functional capacity	[36]
32 premanifest, 48 manifest HD patients, and 24 controls	CSF NF-L	ELISA	Elevated CSF NF-L in premanifest and manifest HD compared to controls, and increased levels in manifest compared to premanifest HD patients	[37]
**Parkinson’s Disease and Parkinsonian Disorders**				
Cohort 1: 171 PD, 30 MSA, 19 PSP, five CBS, and 53 healthy controls. Cohort 2: 20 PD, 30 MSA, 29 PSP, 12 CBS, and 26 healthy controls	sNF-L and pNF-L	Simoa	Elevated sNF-L and pNF-L in MSA, PSP, and CBS vs. PD and healthy controls. sNF-L and pNF-L discerned between PD and MSA, PSP, and CBS	[38]
26 non-demented PD and 23 demented PD patients, and 59 controls	pNF-L	Simoa	Elevated pNF-L in demented vs. non-demented and controls. pNF-L associated with cognition	[31]
64 non-demented PD patients and 21 controls	CSF NF-L	ELISA	CSF NF-L levels increased in PD patients over 2 years, but not in controls	[39]
68 PD, 34 MSA, 34 PSP, and 15 CBS	CSF NF-L	ELISA	CSF NF-L associated with increased mortality	[40]
**Stroke—AIS, TIA and HS**				
595 AIS enrolled within 14 days after symptom onset and 600 controls	sNF-L	Simoa	sNF-L levels highest 3 months post-stroke. sNF-L associated with stroke severity and poor outcomes. sNF-L levels higher compared to controls.	[41]
101 AIS and 35 TIA patients enrolled within 1–12 days (63.8 ± 50.1 h)	sNF-L	Simoa	Elevated sNF-L in AIS compared to TIA. sNF-L correlated with final infarct volume.	[42]
44 controls and 54 AIS on day 1, week 1, and 3–6 weeks post-stroke	s-pNF-H	ELISA	Elevated s-pNF-H in AIS vs. controls. s-pNF-H at week 3 correlated to stroke severity, size, and outcome	[43]
30 controls and 196 AIS patients on admission, days 2, 3, and 7, as well as 3 and 6 weeks post-stroke	sNF-L	Simoa	Elevated sNF-L at admission until 6 months in AIS compared to controls. sNF-L correlated with infarct volume at day 7	[44]
504 AIS and 111 TIA patients within 24 h of symptom onset	sNF-L	ECL	Elevated sNF-L in AIS vs. TIA. sNF-L associated with NIHSS and TIA diagnosis but not infarct size or functional outcome at 3 months	[45]
20 controls and 20 AIS enrolled within 5–10 days	CSF NF-L	ELISA	Elevated NF-L in AIS compared to controls. NF-L correlated with the degree of WML	[46]
49 CeAD patients (10 TIA, 31 AIS, eight local symptoms) within 30 days of symptom onset	sNF-L	ECL	Elevated sNF-L in CeAD stroke vs. CeAD TIA. SNF-L associated with NIHSS. Elevated sNF-H levels within 24 h post-stroke	[47]
22 AIS patients enrolled within 6–24 h after symptom onset	sNF-H	ELISA	Elevated sNF-L levels in RSSI at baseline and 3 months vs. controls. sNF-L associated with RSSI size and baseline WMH severity	[48]
79 RSSI at baseline, 3 and 15 months post-stroke and 53 community-dwelling healthy controls with comparable WMH	sNF-L	Simoa	NF-M levels higher in HS vs. AIS and controls	[49]
10 controls, 11 AIS, and 30 HS within 1 day (TIA) and 5 days (HS)	CSF NF-M	ELISA	NF-H correlated with functional outcome at discharge	[50]
20 controls and 33 AIS patients within 3 days of symptom onset	CSF NF-H and sNF-L	ELISA	No difference between AIS and controls	[51]
**Stroke—Small Vessel Disease**				
53 SVD patients	CSF NF-L	ELISA	CSF NF-L associated with volume of WMLs	[52]
93 controls, 53 CADASIL, and 439 SVD patients	sNF-L	Simoa	Elevated sNF-L in CADASIL and SVD patients compared to controls. sNF-L levels associated with imaging and clinical features of SVD	[53]
**Stroke-SAH**				
35 aSAH patients serially for up to 15 days	CSF NF-L	ELISA	Elevated CSF NF-L levels. No effect on secondary adverse events or long-term outcome	[54]
40 patients with Fisher grade 3 hemorrhage within 6 hrs each day for up to 8–12 days	CSF pNF-H, s-pNF-H	ELISA	pNF-H levels differentiated between patients with poor and favorable outcomes	[55]
**Traumatic Axonal Injury**				
**Severe Traumatic Brain Injury**				
172 neurocritical TBI at bout and at 12 months	sNF-L	Simoa	sNF-L increased over 2 weeks, predicted 12 months outcome	[56]
182 TBI, outcome 6–12 months	CSF NF-L, sNF-L	ELISA	Higher NF-L CSF and serum levels correlated to GCS and predicted a poor clinical outcome	[57]
72 TBI and 35 controls	sNF-L	Simoa	High initial NF-L levels predicted poor clinical outcome at 1 year	[58]
**Mild Traumatic Injury**				
107 mild traumatic brain injury	pNF-L	Simoa	Early levels of NF-L predicted outcome 6–12 months	[59]
118 elderly mild traumatic injury +/- neurological disorders + 40 age-matched controls	sNF-L	Simoa	Older age and neurological diseases are associated with elevated serum NF-L levels in TBI and controls	[60]
**Sport—Related Concussion**				
142 American football + hockey players with concussion: preseason, day 6 and 14	sNF-L	Simoa	No difference in sNF-L with uncomplicated concussion	[61]
45 boxers and 25 controls	sNF-L, CSF NF-L	ELISA	Higher NF-L CSF and serum levels correlated to GCS and predicted a poor clinical outcome. Elevated CSF NF-L after bout and after 14 days	[62]
Experimental soccer headings vs. sham vs. control	pNF-L	ELISA	Elevated p NF-L at 1 h and 1 month	[61]
19 American football players, 19 swimmers, eight samples over 6 months	sNF-L	Simoa	Increased sNF-L over time in football players	[63]
18 football players, accelerometer-embedded mouthguard	pNF-L	Simoa	The frequency and magnitude of head impacts associated with increased NF-L levels	[64]
14 boxers, 35 hockey players, 26 controls, bout +3 months	sNF-L	Simoa	High NF-L levels after bout, returned to normal after 3 mths + higher NF-L levels predicted longer post-concussion symptoms	[58]
**Spinal Cord Injury**				
23 SCI (six cervical fracture, 17 whiplash) and 24 controls	CSF NF-L	ELISA	NF-L increased in proportion to neurological deficits	[65]
27 SCI and 67 controls	sNF-L	Simoa	Serum NF-L concentrations in SCI patients closely correlated with acute severity and long-term outcome	[66]
**Cardiac Arrest**				
22 CA patients 2 or 3 weeks post-CA	CSF NF-L	ELISA	CSF NF-L levels were a reliable measure of brain damage and predictive of poor outcome	[67]
21 CA patients 2 weeks post-CA and 21 controls	CSF NF-L	ELISA	CSF NF-L levels increased in CA patients vs. controls. CSF NF-L levels highest in CA patients with poor outcome	[68]
14 CA patients within 17 days post-CA	sNF-L	Simoa	sNF-L levels associated with time to return of spontaneous circulation and brain damage. sNF-L levels were higher among CA patients who had died vs. CA patients alive at 1 months post-CA	[69]
26 ROSC CA and 26 non-ROSC CA patients on admission	s-pNF-H	ELISA	No difference in pNF-H levels between ROSC and non-ROSC CA patients	[70]
90 CA patients treated with hypothermia sampled over a period of 72 h after CA	p-pNF-H	ELISA	p-pNF-H levels were higher at 2 and 36 h after CA in patients with poor outcome vs. those with good outcome. p-pNF-H levels correlated to neurological prognosis	[71]
717 CA patients sampled three times within the first 72 h after CA	sNF-L	Simoa	sNF-L levels were higher in patients with poor neurological outcome vs. those with good outcome	[72]

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
