# Peer review of "Neurofilaments: The C-Reactive Protein of Neurology"

_brainsci, 2020, doi:10.3390/brainsci10010056_

Round 1

Reviewer 1 Report

I've red the review article from Lambertsen et al. on "Neurofilaments: the C reactive protein of neurology" with interest. 

The idea to evaluate available literature on the role of neurofilaments in neurological diseases is good, but there are several concerns that have to be mentioned here.

First, there are several mistakes in the manuscript. It starts with the title "...the C reactive protein of of...". Second, the headline for section 2 is completely missing.

English and grammar need to be significantly improved by a native speaker.

The abstract is poorly written and does not give an appropriate overview about the content of the manuscript.

The aim of the review was to report on strengths and pitfalls of NFs, but exactly this is mainly missing in the manuscript. The authors summarize a lot of literature, but the evaluation of available studies is at least on the surface. Some mistakes in the manuscript give the impression that the authors were not very accurate by writing the manuscript or by proofreading; e.g. in line 27 it is reported on "cerebrovascular fluid (CSF)". The correct name is cerebrospinal fluid (CSF).

In section 2 the authors report on several neurological diseases, but it is more a summary of literature. This review would profit from a well-balanced evaluation of the studies. The authors need to work out the strengths and pitfalls, which is missing in the majority of diseases.

Table 1 is not very well structured and is confusing.

From the reviewer's point of view the discussion and conclusion section need to be heavily improved to make this manuscript a review article.

Based on the above mentioned concerns my recommendation to the editor is: major revision.

Author Response

Response to reviewer #1

We thank the reviewer for the constructive comments and suggestions to improve our manuscript. Please see below the point by point replies to the points raised by the reviewer.

#1: English and grammar need to be significantly improved by a native speaker.

We apologize for this and have now had the manuscript proof-read and corrected by a native speaker. We hope this has eliminated any typos and improved grammar.

#2: Included a heading for section 2

We have included a heading for section 2 and divided the manuscript more clearly into sections of Introduction, Discussion, and Conclusion.

#3: Improved the abstract to give an overview of the content

We have revised the abstract to better reflect the content of the manuscript.

#4: Evaluation of studies including strengths and pitfalls

We have included a paragraph in each section describing the strength and weaknesses of the studies in the context of the specific disease.

#5: Table 1 is not very well structured and is confusing.          

We agree with the reviewer and have now modified Table 1. We hope the reviewer finds the Table structured now.

Reviewer 2 Report

This is an exhaustive review covering most important studies on NF-L as a biomarker for neurological diseases. Conclusions are excellent. I have the following minor comments:

Please correct the title (that I really like!): there are two “of” in it. Simoa for measuring NF-L was not introduced 2 years ago, as stated in the last sentence of page 1/first sentence of page 2. The first Simoa-based method for NF-L was Gisslén M et al., EBioMedicine. 2015 Nov 22;3:135-140. The Simoa technique itself was first detailed in 2010: Rissin DM et al., Nat Biotechnol. 2010 Jun;28(6):595-9. In paragraph 2.7, it is stated that “blood NF-L does not reach detectable levels in PD”. Using high sensitivity assays, it is possible to quantify NF-L in the blood of any human being. It would be more correct to state that NF-L concentration is not increased in PD compared with control individuals. In paragraph 2.8: Please consider including and commenting on Pedersen A et al., J Neurol. 2019 Nov;266(11):2796-2806. This study contains rather important data on outcome prediction and the temporal profile of NF-L after stroke. From this study, it is clear that NF-L is a rather slow marker, which is important to consider in acute conditions. Please consider including a paragraph on plasma/serum NF-L for neurological outcome prediction after cardiac arrest. There is a growing literature on this topic.

Author Response

We thank the reviewer for the constructive comments and suggestions to improve our manuscript. Please see below the point by point replies to the points raised by the reviewer.

Response to reviewer #2

#1: Simoa for measuring NF-L was not introduced 2 years ago, as stated in the last sentence of page 1/first sentence of page 2. The first Simoa-based method for NF-L was Gisslén M et al., EBioMedicine. 2015 Nov 22;3:135-140. The Simoa technique itself was first detailed in 2010: Rissin DM et al., Nat Biotechnol. 2010 Jun;28(6):595-9.

We thank the reviewer for pointing out this mistake. We have now modified this paragraph and included both Gisslén et al., 2015 and Rissin et al., 2010 to acknowledge the correct studies.

#2: In paragraph 2.7, it is stated that “blood NF-L does not reach detectable levels in PD”. Using high sensitivity assays, it is possible to quantify NF-L in the blood of any human being. It would be more correct to state that NF-L concentration is not increased in PD compared with control individuals.

We agree with the reviewer on this point. We have modified the paragraph as suggested.

#3 Please consider including and commenting on Pedersen A et al., J Neurol. 2019 Nov;266(11):2796-2806. This study contains rather important data on outcome prediction and the temporal profile of NF-L after stroke.

As suggested by the reviewer, we have now included Pedersen et al, J Neurol 2019 in both the manuscript and in Table 1.

#4: Please consider including a paragraph on plasma/serum NF-L for neurological outcome prediction after cardiac arrest. There is a growing literature on this topic.

We thank the reviewer for suggesting to include the growing literature on the use of neurofilaments as a biomarker in cardiac arrest.

As suggested by the reviewer, we have included a paragraph on NFs as a marker for neurological outcome prediction after cardiac arrest. We have also included literature on this subject in Table 1.

Round 2

Reviewer 1 Report

From the reviewer's point of view the authors significantly improved the quality of their manuscript. It is now much better readable with regard to english language, structure and content. 

Table 1 and the abstract were improved and are acceptable now.

The sections on the different neurological disease are well organized and the short conclusion of value and pitfalls of neurofilaments in this field of interest at the end of all sections significantly strengthens the quality of the review. This was very well done by the authors! Congratulations!

I welcome the new section on the role of neurofilaments in the field of cardiac arrest. This further improves the quality of this review.

I would definitely recommend to add one more interesting field to this review, which was very recently published in literature. I would suggest to add a section on sepsis-associated encephalopathy and delirium. First papers focused on that field and gave evidence for a potential role of neurofilaments in these conditions. It would perfectly round off this manuscript, especially with a look at the title, which supports a link between (neuro)inflammation and neurofilaments ("... the C-reactive protein of neurology").

Relevant and recently published papers from the reviewer's point of view might be:

PMID: 31530945 PMID: 30677080 PMID: 31802104 PMID: 31574089

I would highly be interested in reading this manuscript after minor revision again!

Author Response

We thank the reviewer for the positive response to our revised review. As requested, we have included a paragraph describing the use of neurofilaments as biomarkers in delirium. We hope the reviewer likes the paragraph and now finds the review acceptable for publication.